# Downstream Link of Vitamin D Pathway with Inflammation Irrespective of Plasma 25OHD3: Hints from Vitamin D-Binding Protein (DBP) and Receptor (VDR) Gene Polymorphisms

**DOI:** 10.3390/biomedicines13020385

**Published:** 2025-02-06

**Authors:** Mai S. Sater, Zainab H. A. Malalla, Muhalab E. Ali, Hayder A. Giha

**Affiliations:** 1Department of Medical Biochemistry, College of Medicine and Health Sciences (CMHS), Arabian Gulf University (AGU), Manama P.O. Box 26671, Bahrain; zainabhm@agu.edu.bh (Z.H.A.M.); muhalabae@agu.edu.bh (M.E.A.); 2Medical Biochemistry and Molecular Biology, Khartoum, Sudan; gehaha2002@yahoo.com

**Keywords:** inflammation, vitamin D receptor, vitamin D-binding protein, polymorphism, biomarkers, Arabs

## Abstract

**Background:** Vitamin D insufficiency/deficiency is a highly prevalent condition worldwide. At the same time, chronic inflammation is a versatile pathophysiological feature and a common correlate of various disorders, including vitamin D deficiency. **Methods:** We investigated the possible association of inflammation with 25-hydroxyvitamin D3 (25OHD3) levels and its down-stream pathway by exploring vitamin D-binding protein (*DBP*) and vitamin D receptor (*VDR*) genes for single-nucleotide polymorphisms (SNPs), in healthy non-elderly Bahraini adults. Plasma levels of 25OHD3 were measured by chemiluminescence, and six SNPs, four in the *GC* gene (rs2282679AC, rs4588CA, rs7041GT, and rs2298849TC) and two in the *VDR* gene (rs731236TC and rs12721377AG) were genotyped by real-time PCR. The concentrations of five inflammatory biomarkers, IL6, IL8, procalcitonin (PCT), TREM1, and uPAR, were measured by ELISA. **Results:** The results showed no association between the 25OHD3 level and any of the inflammatory markers’ levels. However, three tested SNPs were significantly associated with the concentrations of tested biomarkers except for IL6. The TT mutant genotype of rs2298849TC was associated with lower levels of IL8 and higher levels of PCT and TREM1, the AA mutant genotype of rs2282679AC was associated with decreased levels of IL8 (*p* ≤ 0.001) and increased levels of TREM1 (*p* = 0.005), and the GG wild genotype of rs12721377AG was associated with increased levels of 25OHD3 (*p* = 0.026). **Conclusions:** Although chronic inflammation is not associated with the vitamin D system in the blood, it is downstream, as revealed by *DBP* and *VDR* genotyping. Alternatively, DBP and VDR pursue other functions beyond the vitamin D pathway.

## 1. Introduction

Inflammation is the classical immune system’s physiological and pathophysiological response to harmful stimuli [1]. Therefore, several diseases are associated with chronic inflammation due to external or internal insults triggered by various factors, including pathogens, damaged cells, metabolites, allergens, and toxic compounds [2]. Also, chronic low-grade inflammatory states are associated with aging and frailty, irrespective of co-morbidities, and inflammation is thought to be a potential mechanism of frailty [3,4].

Vitamin D is a fat-soluble vitamin, and together with its transporter and receptor proteins has been added to the list of the endocrine systems [5]. Plasma vitamin D (D_3_), which is mostly produced under the skin and usually supplemented by dietary or therapeutic sources, is metabolized to 25(OH)D_3_ in the liver by vitamin D 25-hydroxylase and then further hydroxylated by the 1α-hydroxylase (CYP27B1) in the kidney to the biologically active form, 1,25(OH)_2_D_3_ (calcitriol) [6,7]. The active vitamin D is transported in blood by vitamin D-binding protein (DBP). In the target tissues, the calcitriol binds and activates the vitamin D receptor (VDR), a member of the superfamily of nuclear receptors, and functions as a ligand-activated transcription factor [8]. The DBP is encoded by the group-specific component gene (*GC*). Besides transporting vitamin D metabolites, DBP has other functions, including the modulation of immune and inflammatory responses, binding of fatty acids, and controlling bone development [9]. Age, sex, and obesity are expected to affect or be affected by DBP levels. High DBP has been found to be associated with obesity in females [10], in contrast to other studies that showed that DBP levels are not associated with obesity [11], or differed significantly between healthy men and women [12]. This controversy is suggested to be due to increased inflammatory response in obese women [13]. However, there have been no reports on the association of DBP with age. Regarding the VDR, is worth noting that vitamin D has both nuclear and cytoplasmic actions [14]. In the nucleus, gene expression in target tissues is regulated by calcitriol via a specific cytosolic/nuclear VDR protein, a member of the steroid/thyroid hormone-receptor superfamily [15]. Also, vitamin D acts via a non-genomic pathway through the putative membrane vitamin D receptor (mVDR), which might be responsible for vitamin D’s rapid effects [16]. However, the phenotypic effects of VDR gene polymorphisms on inflammation and other disorders have been under-reported [17].

Several studies have linked vitamin D insufficiency/deficiency with chronic inflammation [18,19]. Also, vitamin D is associated with several metabolic disorders, including diabetes, insulin resistance, and obesity [7,20], which are themselves associated with chronic metabolic inflammation [21,22]. Vitamin D possibly regulates the adaptive immune response in different inflammatory disorders [23,24] and modulates innate immune responses to various pathogens [25].

Numerous biomarkers are used to confirm inflammation; some are specific and others are versatile [26]. There are two main families of biomarkers of inflammation: the circulating cytokines, such as interferons (IF) and interleukins (IL), e.g., IL-1β, IL-6, IL-8, IFγ, and TNF-α, many of which are, however, not routinely used in clinical practice [27]. The second family, secreted by the liver, belongs to the acute phase proteins (APPs), such as C-reactive protein (CRP), haptoglobin, or serum amyloid a (SAA), which is well correlated with clinical outcomes and diseases, such as metabolic disorders [26]. However, an expanding third group of inflammatory biomarkers is regularly updated with new members. This group is not limited to a specific family of molecules; it includes biomarkers such as procalcitonin (PCT), TREM-1, uPAR, and presepsin, which have showed promising results [22].

This study examined the possible association of inflammation with plasma vitamin D concentration and polymorphisms in genes encoding DBP (chromosome 4) and VDR (chromosome 12) proteins, with correction for the major confounders, the age, sex, and co-morbidities, in native Bahraini Arabs.

## 2. Materials and Methods

### 2.1. Study Site, Design, and Subjects

The current cross-sectional prospective study was carried out in the Bahrain Defense Force (BDF) hospital. The study subjects were seemingly healthy, clinically asymptomatic, and were unknown to have a chronic disease or severe acute disorder. They were voluntarily recruited during their regular medical checkups or blood donations at the clinical chemistry laboratory and blood bank in BDF, respectively. The participants were Bahraini Arabs, while non-Arab or naturalized Bahraini, pregnant or lactating females, and patients known to have acute or chronic symptomatic illness including diabetics, were excluded. A total of 162 subjects (76 males and 86 females), aged between 20 and 40 years, with BMI evenly distributed over the spectra of obesity, normal (N, n = 43), overweight (OW, n = 41), obese (OB, n = 39), and severely obese (SOB, n = 39), were selected (Table 1). The participants’ demographic and biochemical profiles are shown in (Table 1).

### 2.2. Ethical Issues

This study was approved by the Arabian Gulf University (AGU), College of Medicine and Health Sciences (CMHS), and BDF Hospital research and ethics committees, as part of a Master’s in Laboratory Medicine (MLM) student project. Informed consents were obtained from all study subjects before inclusion in the study and blood sampling.

### 2.3. Blood Sample Collection

The blood samples were collected from all the donors during the regular blood drawings for routine workouts on predetermined dates, after overnight fasting (10–12 h). Plain tubes and EDTA-coated tubes were used for blood collection by vein puncture. Following blood centrifugation, the buffy coat was stored in liquid nitrogen and freezers at −80 °C for later DNA extraction. The plasma was transferred into plain tubes, acidified with 1/10 volume of 1 M HCl solution, and stored at −80 °C until use, and the serum was stored at −20 °C.

### 2.4. Measurements of Plasma Vitamin D and Inflammatory Biomarker Levels

#### 2.4.1. Chemiluminescent Microparticle Immunoassay (CMIA)

We measured serum 25-hydroxyvitamin D (25OHD3) concentration, the primary circulating form of vitamin D, by chemiluminescent microparticle immunoassay (CMIA), (Architect Abbott Diagnostics, Lake Forest, IL, USA), using an Abbott Architect analyzer at the Salmaniya Medical Complex (SMC), according to the manufacturer’s instructions. Six calibrators were used to standardize the assay, and two controls (low and high) were used to monitor Internal Quality Assurance—IQA. For more details, a specific antibody for 25OHD3 was used to coat the magnetic microparticles (solid phase), while 25OHD3 was linked to anti-biotin acridinium conjugate. During the incubation, 25OHD3 dissociated from its binding protein and competed with the labeled one for the binding sites on the antibody. After incubation, the unbound material was removed with a wash cycle. Subsequently, the starter reagents were added and a flash chemiluminescent reaction was initiated. The light signal was measured and plasma 25OHD3 levels were estimated in nmol/L [20].

#### 2.4.2. Enzyme-Linked Immunosorbent Assay (ELISA)

The plasma concentration of the tested cytokines and inflammatory markers were measured by solid-phase sandwich ELISA, using Invitrogen ELISA kits, for IL-6 (EH2IL6), IL-8 (KHC0081), PCT (EHPCT), TREM1 (EHTREM1), and uPAR (EHPLAUR), following the kits’ protocols, as described before [22]. Thermo Multiscan Spectrum Plate Reader was used with SkanIt RE for MSS 2.4.2 software, for the plates’ absorbance measurement. The standard curve for each tested marker was generated by plotting the Absorbance (Abs) of the standards against their known concentrations in pg/mL (Appendix A).

### 2.5. Genotyping of Vitamin D Receptor (VDR) and Vitamin D-Binding Protein (GC) Genes

#### 2.5.1. DNA Extraction

The leukocyte buffy coat was used to extract the total genomic DNA, using a QIAamp DNA Blood Mini Kit 250 (Qiagen, Venlo, The Netherlands), according to the manufacturer’s protocol.

#### 2.5.2. The Tested SNPs of DBP and VDR Genes

Previously tested SNPs of *DBP* (chromosome 4) and *VDR* (chromosome 12) genes for the study of obesity in the same subjects [28] were used in the current study. The four *DBP* gene SNPs are as follows: an intron transversion substitution rs2282679 AC, intron transition substitution rs2298849 TC, and missense transversion substitution rs4588 CA and rs7041 GT. The *VDR* gene SNPs were a silent transition substitution rs731236 TC, and an intron transition substitution rs12721377 AG.

#### 2.5.3. Real-Time PCR Analysis

Real-time polymerase chain reaction (rtPCR) was deployed for *VDR* and *GC* genotyping using the Applied Biosystem (ABI) StepOne PCR System, which is based on the fluorescent signal detection, during each PCR cycle. TaqMan^®^ pre-designed SNP genotyping assay kit was used for each allele. The method was carried out in two steps: PCR amplification and the endpoint detection of fluorescent signals. The allelic discrimination was completed by selective annealing of TaqMan^®^ MGB probes [28].

### 2.6. Statistical Analysis

The statistical analysis was carried out using Sigma Stat software (Systat Software Inc., version 3.5. Copyright 2006). Mann–Whitney Rank Sum Test (MW), Kruskal–Wallis One Way Analysis of Variance on Ranks (KW) tests, and All Pairwise Multiple Comparison Procedures (Dunn’s Method) were used for comparisons between the study groups. For correlation analysis, we used Pearson Product Moment Correlation. The statistical significance was set at *p* < 0.05. For more stringent comparisons of the two principal variables, the 25OHD3 and biomarkers, we used quartiles, which are types of quantiles that divide the number of data points after ranking into four quarters (1st Q, 2nd Q, 3rd Q, and 4th Q) of more-or-less equal size.

## 3. Results

### 3.1. Description of Study Subjects

The total number of the study subjects was 162 (76 males and 86 females), with a mean (±SD) age of 29.99 ± 5.65 years. The subjects had a mean weight of 84.61 ± 23.65 kg; however, they were evenly distributed over the obesity scales, with a mean (±SD) BMI of 30.46 ± 7.921 kg/m^2^. The average glycemic and lipid profiles were all within reference ranges (Table 1).

### 3.2. Correlation of Plasma Concentrations of Vitamin D with Inflammatory Biomarkers

The plasma 25OHD3 levels were not correlated with the levels of the tested inflammatory biomarkers: IL-6 (*p* = 0.416, CC-0.0728), IL-8 (*p* = 0.290, CC-0.0946), PCT (*p* = 0.554, CC-0.0530), TREM-1 (*p* = 0.452, CC-0.0673), and uPAR (*p* = 0.574, CC-0.0521), (Figure 1).

### 3.3. Comparisons of the 25OHD3 Concentrations Between Quartiles of Each Inflammatory Biomarker

There were significantly marked differences in the concentrations of inflammatory biomarkers (Il-6, IL-8, PCT, TREM-1, and uPAR) between the quartiles (Q1st–Q4th) of each biomarker, with *p* < 0.001 for each comparison (Figure 2A-i–E-i), KW test. However, the concentrations of 25OHD3 were comparable between the quartiles of each inflammatory biomarker; IL-6 (*p* = 0.381), IL-8 (*p* = 0.530), PCT (*p* = 0.622), TREM-1 (*p* = 0.862), and uPAR (*p* = 0.578) (Figure 2A-ii–E-ii).

### 3.4. Comparisons of the Concentrations of the Inflammatory Biomarkers in the Different Quartiles of 25OHD3 Concentrations

As seen in (Table 2), 127 subjects who had their vitamin D measured were distributed into four groups (quartiles—Q) based on their plasma 25OHD3 concentration. The levels of the tested inflammatory biomarkers, IL-6, IL-8, PCT, TREM-1, and uPAR, were not significantly different between the different quartiles of the 25OHD3 concentrations; *p*-values were 0.422, 0.895, 0.740, 0.236 and 0.830, respectively, KW test.

### 3.5. Association of Inflammatory Biomarker Levels with Vitamin D-Binding Protein (DBP) and Vitamin D Receptors (VDR) SNP

As seen in Table 3, the IL-6 levels were comparable between the genotypes of the SNPs rs2282679 AC (*p* = 0.463), rs4588 CA (*p* = 0.089), rs7041 GT (*p* = 0.794), and rs2298849 TC (*p* = 0.699) of the *DBP* gene. Similarly, the IL-6 levels were comparable between the genotypes of the SNPs rs731236 TC (*p* = 0.231) and rs12721377 AG (*p* = 0.389) of the *VDR* gene.

The IL-8 levels were significantly different between the genotypes of the SNP rs2282679 AC, with markedly lower levels in the carriers of minor (mutant) genotype AA (*p* < 0.001), and of the mutant genotype (TT) of rs2298849 TC (*p* < 0.001) SNPs of the *DBP* gene. The levels of IL-8 were comparable between the genotypes of the SNPs rs4588 CA (*p* = 0.925) and rs7041 GT (*p* = 0.766). Similarly, the IL-8 levels were comparable between the genotypes of the SNPs rs731236 TC (*p* = 0.307) and rs12721377 AG (*p* = 0.601) of the *VDR* gene.

The PCT levels were comparable between the genotypes of the SNPs rs2282679 AC (*p* = 0.388), rs4588 CA (*p* = 0.443), and rs7041 GT (*p* = 0.698). In contrast, the PCT levels significantly differed between the genotypes of the SNPs rs2298849 TC (*p* = 0.011) of the *DBP* gene, with markedly higher PCT levels recognized in subjects carrying the mutant genotype TT. However, the PCT levels were comparable between the genotypes of the rs731236 TC (*p* = 0.765) and rs12721377 AG (*p* = 0.965) SNPs of the *VDR* gene.

The TREM-1 levels were significantly different between the genotypes of the *DBP* gene SNP rs2282679 AC, with higher levels in the carriers of mutant genotype AA (*p* = 0.005), and of the mutant genotype (TT) of rs2298849 TC (*p* = 0.012). The levels of TREM-1 were comparable between the genotypes of the SNPs rs4588 CA (*p* = 0.708) and rs7041 GT (*p* = 0.880). Similarly, the TREM-1 levels were comparable between the genotypes of the SNPs rs731236 TC (*p* = 0.077) and rs12721377 AG (*p* = 0.191) of the *VDR* gene.

The uPAR levels were comparable between the genotypes of the tested SNPs in the *DBP* gene: rs2282679 AC (*p* = 0.858), rs4588 CA (*p* = 0.506), rs7041 GT (*p* = 0.729), and rs2298849 TC (*p* = 0.350). Similarly, the uPAR levels were comparable between the genotypes of the SNP of the *VDR* gene, rs731236 TC (*p* = 0.397); however, the levels of uPAR were significantly higher in subjects carrying the major genotype GG of the SNP rs12721377 AG (*p* = 0.026). The mutant AA genotype of rs12721377 AG was recognized in only two subjects and, therefore, was excluded from the analysis.

## 4. Discussion

In this study, vitamin D concentration was not associated with inflammation in middle-aged (20–40 years) asymptomatic seemingly healthy adults. This is indicated by the lack of correlations between the plasma levels of all tested inflammatory biomarkers with the plasma concentration of vitamin D, and the comparable levels of the tested biomarkers between subjects in the different quartiles of the plasma vitamin D levels. Interestingly, while vitamin D levels were not associated with inflammation, the polymorphisms of the genes in the vitamin D pathway, namely *DBP* (*GC*) and *VDR* genes, were found to influence the levels of the inflammatory biomarkers markedly.

Three major potential confounders might intervene with the interpretation of the results regarding the association between inflammation and vitamin D as dependent or independent variables: age [4,29], sex [28,30], and co-morbidities [18,21]. Importantly, the three of them were corrected by default in this study by the study design, where the effect of aging was overcome by limiting the study subjects’ ages to 20–40 years, and the sex was similar in the comparison groups (Table 1). Also, the influence of the diseases/disorders that trigger inflammatory responses was lessened to the minimum by limiting the study to apparently healthy subjects.

The present study found no association between inflammation and plasma vitamin D concentration in non-elderly healthy subjects. A low vitamin D level was previously reported to be associated with inflammation [31]. However, the subjects of the previous study were hospitalized patients; in such a situation, the cause of hospitalization could be the cause of inflammation. Another large-scale community study in Ireland has found that older adults with vitamin D deficiency have higher levels of inflammation markers than those with sufficient levels of the vitamin [29]. However, age on its own is a cause of chronic inflammation; also, the diseases of aging might be contributors to the inflammation; therefore, both confounders weaken this link between vitamin D and inflammation in the Laird et al. study [29]. To our knowledge, no study has reported the association of inflammation with vitamin D system disorders after correction for the major confounders mentioned above.

Interestingly, vitamin D-attributed disorders may occur on the background of normal plasma vitamin D levels. This is suggesting that molecular defects downstream of the vitamin D pathway, including vitamin D-binding proteins and receptors, and signal transduction pathways together with intranuclear DNA elements, are possibly involved in disease development [32]. In the present study, six polymorphisms in the *DBP* and *VDR* genes were tested, and three of them were found to be associated with inflammation, as indicated by the significantly raised/reduced concentrations of the inflammatory biomarkers among the genotype carriers (Table 3). The AA mutant genotype of the *DBP* gene, rs2282679, was associated with markedly low levels of IL-8; on the contrary, it was associated with significantly high levels of TREM-1. In this setting, we previously reported that this SNP minor allele (A) was associated with low 25OHD3 plasma levels, but not with BMI [28]. On the contrary, this SNP was not associated with rheumatoid arthritis or systemic lupus erythematous, the typical inflammatory and immunological disorders [33]. In another study, rs2282679 was positively associated with TNF-α levels; however, the study was conducted on children with T1D [34]. The second SNP in the *DBP* gene that was found to be associated with inflammation was rs2298849TC. The TT mutant genotype of this SNP was similarly associated with low IL-8 and high TREM-1; in addition, it was associated with markedly high PCT levels. In this setting, the rs2298849TC SNP was not associated with obesity or 25OHD3 levels [28]. In another study, the AA genotype was found to be associated with decreased serum levels of calcitriol (active vitamin D) in high-risk coronary artery disease (CAD) patients [35]. In the present study, the third SNP associated with inflammation was rs12721377AG in the *VDR* gene, where the GG wild genotype was associated with inflammation, as indicated by the high levels of uPAR. For this SNP, only 2 subjects were found to carry the AA mutant genotype, and about 20 subjects were mutant allele (A) carriers with a heterozygous AG genotype. Previously, we showed that the AA genotype of this SNP was associated with low 25OHD3 levels in females [28]. Upon literature review, it was seen that the rs12721377AG SNP has rarely been investigated before [36].

Although the remaining three SNPs were not associated with inflammation in the current study, they have been in other studies. For example, a review by Malik et al. reported that rs7041 GG and rs4588 CC were common modulators of immunity and possibly play a role in protracted innate immune-related inflammation in response to tissue injury or repeated infection [37]. We previously found that both SNPs were associated with high BMI in females and that the latter SNP was also associated with low 25OHD3 levels in males [28]. The third SNP, rs731236 (G), was found to be associated with obesity and the up-regulation of inflammasome components, mainly the proinflammatory cytokines [38].

Furthermore, the *DBP* gene polymorphisms have specific allele distributions in different geographic areas, with consequent ethnicity-specific effects on DBP levels and vitamin D-binding affinity and disease susceptibility [9,39]. The present study was carried out in a different ethnic group, which represents the Arabs in the Middle East, where vitamin D insufficiency/deficiency was markedly high (>95%) [20].

Finally, although the subjects in this study were healthy and asymptomatic, or at least were not known to have any chronic disorder, we observed marked individual variations in the levels of each of the tested inflammatory biomarkers (Figure 2), with a significant difference between each quartile and others for each biomarker, which cannot be attributed to any of the known confounders. The second observation was that there were no correlations between the concentrations of any two of the tested inflammatory biomarkers. The two observations indicate that each inflammatory biomarker is triggered by different stimulants and is released independently of the others. The raised level of a single biomarker does not necessarily imply a patent disease. Moreover, correction of the routinely measured plasma 25OHD3 for *DBP* and *VDR* abnormalities, and likely other downstream polymorphisms, to estimate the ‘effective D3’ instead of the ‘active D3’ (calcitriol) might be a more precise estimation of the functional vitamin D.

The strengths of this study are as follows: it is probably the first to address the issue of the association of inflammation with vitamin D system disorders in isolation from the most confounding factors of inflammation, by correcting for age, sex, and co-morbidities. Moreover, the study was carried out in a culturally and ethnically different setting from the ones in which similar studies were conducted, thus filling a gap in the literature. The limitations of this study are the relatively small sample size of the study subjects, and the limited number and diversity of the tested inflammatory biomarkers and diagnostic investigations. The latter is needed for screening the study subjects for the largest number of disorders to unravel the probable hidden or asymptomatic diseases which might elicit inflammatory response. Therefore, more studies in different settings are required to address the above shortcomings. In conclusion, this study showed a link between chronic inflammation and the polymorphism of the *DBP* and *VDR* genes, independent of the plasma vitamin D level, after correction for age, sex, and co-morbidities. In addition, it contributes to a better understanding of the pathophysiology of inflammation by exploring the roles of the different components of the vitamin D pathway in this disorder.

## Figures and Tables

**Figure 1 biomedicines-13-00385-f001:**
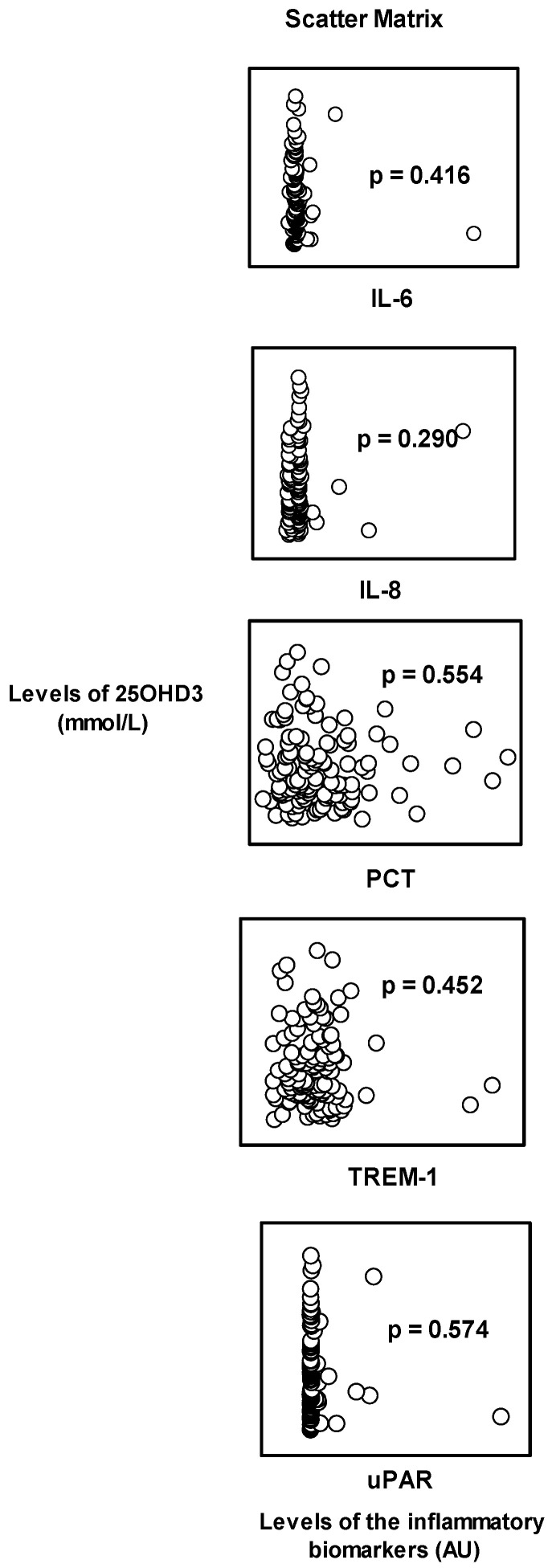
Correlations of the plasma concentrations of vitamin D (25OHD3), in samples collected from healthy Bahrani subjects aged 20–40 years, with the levels of each tested inflammatory biomarker (IL-6, IL-8, PCT, TREM-1, and uPAR). The Pearson Product Moment Correlation test was used for statistical analysis. No significant correlation was observed; the *p*-values are shown in the figure.

**Figure 2 biomedicines-13-00385-f002:**
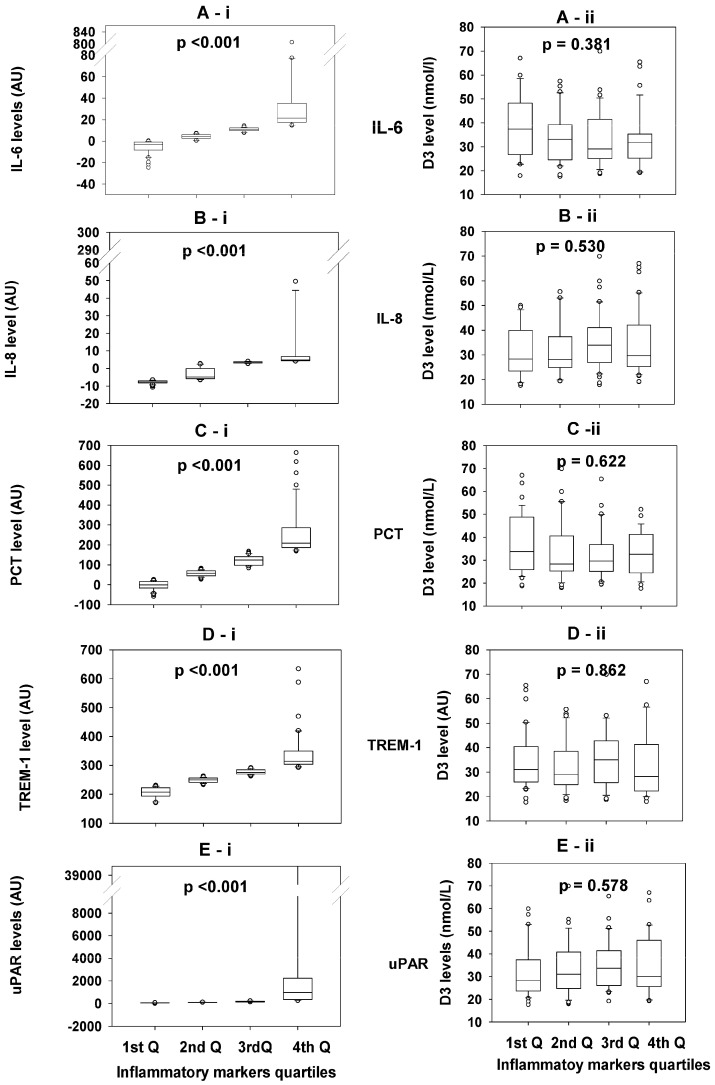
The figure shows the concentration quartiles of the tested inflammatory biomarkers (IL-6, IL-8, PCT, TREM-1, and uPAR); the left-side panel (**A-i**–**E-i**) demonstrates significantly markedly different levels between the quartiles of each biomarker, with *p*-values for each <0.001. The right-side panel (**A-ii**–**E-ii**) shows comparable concentrations of 25OHD3 between the quartiles of each tested biomarker. Kruskal–Wallis One Way Analysis of Variance on Ranks (KW) was used for statistical analysis, with *p*-values for all >0.05, as shown in the figure.

**Table 1 biomedicines-13-00385-t001:** Description of study subjects: demographic, lipid, and glycemic profiles.

Parameters	Values
Number	162
Age (years)	29.99 ± 5.65
Sex (M/F)	76/86
BMI (kg/m^2^)	30.46 ± 7.921
Weight (kg)	84.61 ± 23.65
Obesity grades [kg/m^2^]	
Normal [18.5 to 24.9]	43
Overweight [25 to 29.9]	41
Obese [30 to 34.9]	39
Severely obese [≥35]	39
Lipid profile	
TC (mmol/L)	4.94 ± 1.04
LDL-c (mmol/L)	3.02 ± 0.96
HDL-c (mmol/L)	1.19 ± 0.34
TAG (mmol/L)	1.43 ± 1.29
Glycemic profile	
FBG (mmol/L)	5.54 ± 1.39
HbA1c (mmol/mol)	36.75 ± 6.07

**Table 2 biomedicines-13-00385-t002:** Comparisons of the plasma concentration of the inflammatory biomarkers between the study subjects in the four quartiles (1st Q–4th Q) of the 25OHD3 concentration.

	Inflammatory Biomarker Concentrations (Arbitrary Units)				
D3 level (nmol/L)	1st Q (n = 32) (17.6–25.2)	2nd Q (n = 32) (25.3–31.7)	3rd Q (n = 32) (31.8–41.1)	4th Q (n = 31) (41.3–69.9)	*p*-value
IL-6	10.52, 2.84–14.56	10.72, 5.66–15.00	8.10, 3.64–18.79	6.83, 0.31–11.84	0.422
IL-8	3.131, −5.670–4.756	3.399, −6.769–4.282	3.295, −3.119–4.093	3.363, −5.287–4.643	0.895
PCT	105.18, 30.44–172.66	65.10, 27.13–150.01	83.17, 16.82–178.61	58.96, 13.05–136.17	0.740
TREM-1	262.76, 234.87–296.64	239.45, 214.21–270.04	254.97, 223.48–287.51	256.46, 222.00–281.89	0.236
uPAR	114.35, 81.92–209.13	116.09, 83.70–282.52	127.07, 89.42–185.52	143.17, 98.37–243.76	0.830

Note: the concentrations of 25OHD3 (D3) are in mmol/L (shown in the 2nd row), while the concentrations of inflammatory biomarkers are in arbitrary units (AU) (shown on the top row).

**Table 3 biomedicines-13-00385-t003:** Comparisons of plasma inflammatory biomarker levels between subjects carrying different genotype variants of vitamin D receptor (*VDR*) and vitamin D-binding protein (*GC*) SNPs.

Levels of	IL-6 (AU)	IL-8 (AU)	PCT (AU)	TREM-1 (AU)	uPAR (AU)
rs2282679 AC vitamin D-binding protein (DBP)					
AA [18]	3.52, −3.48–17.13	−7.07, −8.11–−5.92	137.10, 58.48–203.79	282.12, 267.43–347.81	155.68, 58.55–307.93
AC [63]	9.98, −0.95–15.00	2.69, −6.10–3.85	66.91, 16.50–168.71	262.47, 230.01–296.13	124.06, 87.73–218.78
CC [77]	7.95, 2.57–12.76	3.29, −5.79–4.40	82.76, 40.75–175.00	254.73, 225.17–281.48	128.15, 89.90–226.04
*p*-values	0.463	**<0.001**	0.388	**0.005**	0.858
rs4588 CA (DBP)					
CC [10]	1.66, −5.42–7.76	−1.95, −7.38–4.50	41.21, 19.40–209.91	274.06, 229.55–335.78	102.69, 60.61–223.03
CA [56]	9.07, −0.22–15.06	2.69, −5.64–3.78	76.13, 13.05–161.86	256.77, 230.23–287.62	118.66, 93.16–216.08
AA [91]	8.09, 0.80–14.99	2.87, −6.73–4.22	95.86, 43.48–170.73	263.06, 227.88–291.59	136.92, 84.74–227.58
*p*-values	0.089	0.925	0.443	0.708	0.506
rs7041 GT (DBP)					
GG [35]	3.54, −0.16–12.55	2.20, −6.77–4.33	70.26, 25.69–134.60	273.53, 224.16–294.99	116.09, 93.05–227.58
GT [75]	8.15, −0.24–15.00	2.69, −5.81–4.05	93.78, 18.31–178.10	260.76, 232.77–290.27	129.82, 89.15–251.30
TT [46]	8.191, 0.393–15.102	2.61, −6.98–4.40	96.29, 43.24–177.28	263.11, 225.60–289.82	114.74, 80.85–218.80
*p*-values	0.794	0.766	0.698	0.880	0.729
rs2298849 TC (DBP)					
TT [26]	3.55, −3.16–18.34	−6.74, −7.95-−6.07	135.93, 70.24–173.61	288.71, 248.57–347.47	158.44, 84.39–332.99
TC [64]	7.30, −0.36–15.06	2.69, −5.78–4.11	93.65, 39.86–180.14	258.02, 228.14–283.01	130.18, 92.83–222.23
CC [66]	9.58, 3.56–12.72	3.15, −2.70–4.39	56.03, 13.36–130.50	256.59, 222.89–289.01	114.35, 80.25–182.27
*p*-values	0.699	**<0.001**	**0.011**	**0.012**	0.350
rs731236 TC vitamin D receptor (VDR) gene					
TT [14]	3.76, −1.67–11.83	−4.76, −7.92–4.18	106.73, 56.22–160.11	241.04, 206.19–268.44	107.67, 76.5–136.93
TC [67]	7.59, −0.86–12.89	−0.25, −6.60–4.10	82.76, 28.21–163.70	259.48, 241.10–290.27	119.13, 84.74–218.78
CC [72]	9.58, 2.00–16.02	3.12, −5.66–4.13	79.82, 15.17–177.07	269.03, 226.66–302.32	129.82, 89.90–234.25
*p*-values	0.231	0.307	0.765	0.077	0.397
rs12721377AG (VDR)					
AA [2]	Ex				
AG [20]	6.01, −0.88–11.28	2.93, −6.42–4.26	76.47, 42.68–165.43	269.36, 250.62–293.71	80.06, 62.67–211.56
GG [132]	8.106, 0.54–14.56	2.64, −6.58–4.11	83.17, 23.22–170.23	258.23, 226.66–291.29	128.67, 92.89–229.57
*p*-values	0.389	0.601	0.965	0.191	**0.026**

Note: The numbers between brackets are the number of study subjects. The statistical tests used were as follows: Kruskal–Wallis One Way Analysis of Variance on Ranks (median, 25–75% percentile) was used. AU: arbitrary units. Ex: excluded. Bolded and grey highlighted *p*-values are the only significant ones.

## Data Availability

The data are available from the corresponding authors upon reasonable request.

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
