# Peer review of "Downstream Link of Vitamin D Pathway with Inflammation Irrespective of Plasma 25OHD3: Hints from Vitamin D-Binding Protein (DBP) and Receptor (VDR) Gene Polymorphisms"

_biomedicines, 2025, doi:10.3390/biomedicines13020385_

Round 1
Reviewer 1 Report
Comments and Suggestions for Authors
Title: A Downstream Link of Vitamin D Pathway with Inflammation 2 Irrespective of Plasma 25OHD3: Hints from Vitamin D Binding 3 Protein (DBP) and Receptor (VDR) Gene Polymorphisms
ID: biomedicine-3451782
Journal: biomedicines
The research article investigated the possible association of inflammation with 25-hydroxyvitamin D3 (25OHD3) levels and its down-stream pathway by exploring vitamin D binding protein (DBP) and vitamin D receptor (VDR) genes for single nucleotides polymorphisms (SNPs), in healthy non-elderly-Bahraini-adults. The comments related to this article are given below.
1. In the introduction section, line 55, the authors write, “Low DBP levels are associated 55 with obesity, female sex, and age (10).” Elaborate, how is DBP associated with obesity, female sex, and age, and what is the degree of association? Add a few more references to support this statement.
2. The methodology mentions that the sample collection is carried out in the Bahrain Defense Force (BDF) hospital, in Bahrain. Authors should mention the type of sampling done and hence the bias associated with the study.
3. In the results section, Figure 1., “Correlations of the ---- shown in the figure.” Mention the r2 value for each case to indicate how the model fits the data point.
4. As the inflammatory marker analysis is done using kits. Why have they mentioned the quantifications in arbitrary units and not in terms of concentrations (pg/ml)? Add the standard plot for these estimations.
5. In the A-ii graph, the authors mention D3 in au, whereas in B-ii to E-ii, they write the concentrations in nmoles/L. This isn't easy to understand.
6. How is the data divided into quartiles? Explain this clearly in the manuscript.
7. In the conclusion section authors should identify the translational value of the findings of this study. How can these findings be generalized?
8. Add strengths and weaknesses of this study.
Author Response
Response to reviewers’ comments
Comments and Suggestions for Authors
Title: A Downstream Link of Vitamin D Pathway with Inflammation 2 Irrespective of Plasma 25OHD3: Hints from Vitamin D Binding 3 Protein (DBP) and Receptor (VDR) Gene Polymorphisms
ID: biomedicine-3451782
Journal: biomedicines
The research article investigated the possible association of inflammation with 25-hydroxyvitamin D3 (25OHD3) levels and its down-stream pathway by exploring vitamin D binding protein (DBP) and vitamin D receptor (VDR) genes for single nucleotides polymorphisms (SNPs), in healthy non-elderly-Bahraini-adults. The comments related to this article are given below.
Note: Com. = Reviewer comment; Res. = Our response
- com. In the introduction section, line 55, the authors write, “Low DBP levels are associated with obesity, female sex, and age (10).” Elaborate, how is DBP associated with obesity, female sex, and age, and what is the degree of association? Add a few more references to support this statement.
Res. Based on this valuable comment, we revised the mentioned statement with elaboration on the possible association of these factors with DBP, and added three additional references (PMID: 27169839; PMID: 885987, and, PMID: 31191450)
- Com. The methodology mentions that the sample collection is carried out in the Bahrain Defense Force (BDF) hospital, in Bahrain. Authors should mention the type of sampling done and hence the bias associated with the study.
Res. Although the comment is not clear, we assumed that what is missing for ‘blood sample’, and a possible cause of bias was having two types of donors, accordingly we responded. Under the sub-title Blood sample collection: The samples collected were ‘fasting blood samples’, and were collected from both groups of donors, the volunteers (Blood bank) and those on regular checkups (clinic), in the same way, e.g., timing, fasting.
- Com. In the results section, Figure 1., “Correlations of the ---- shown in the figure.” Mention the r2 value for each case to indicate how the model fits the data point.
Res. Because the p-values were all not significant, we are satisfied with the presentation of the p-values only in the figures' captions, the correlation coefficient (CC or r2) is shown in the text (Result section, subtitle, 3.2). Adding the CC to the figure caption might cause some confusion and will be of no additional value.
- Com. As the inflammatory marker analysis is done using kits. Why have they mentioned the quantifications in arbitrary units and not in terms of concentrations (pg/ml)? Add the standard plot for these estimations.
Res. This study aims to compare the groups and correlate variants for individuals, rather than to determine the actual values, which of no additional value, on the contrary, some values can read as zero after correction which does not suit the correlation analysis. Also, the outcome of the analysis will be the same, whether we used the arbitrary unit (AU), the corrected values, or the pg/ml, since the latter are derived from the AU using constant factors. Also ELISA is semi-quantitative and the results vary from lab to lab, kit to kit, setting to another. The pg/ml will be appropriate if cut-off level and reference ranges for diagnostic purposes are needed. However, the data and the standard plots are available if needed.
- Com. In the A-ii graph, the authors mention D3 in au, whereas in B-ii to E-ii, they write the concentrations in nmoles/L. This isn't easy to understand.
Res. This is by mistake. It is now corrected
- Com. How is the data divided into quartiles? Explain this clearly in the manuscript.
Res. An explanation is added under the statistical analysis section (2.6) of material and methods.
- Com. In the conclusion section authors should identify the translational value of the findings of this study. How can these findings be generalized?
Res. We elaborated a bit in the conclusion to explain the value of the findings
- Com. Add strengths and weaknesses of this study.
Res. At the bottom and before the conclusion we mentioned the strengths and weaknesses plus the limitations and suggestions.
Reviewer 2 Report
Comments and Suggestions for Authors
The paper presented by Sater and colleagues explores the levels of monohydroxy vitamin D and its receptor and analyzes 6 different SNPs in healthy non-elderly-Bahraini-adults. The work is quite interesting but presents some critical issues
The sample size is adequate and well differentiated between normal patients and patients with different degrees of obesity
However, only patients with an age range of around 29.99±5.65 were selected as shown in table 1
The first question concerns the age of the population taken into consideration: vitamin D deficiency is usually associated with a greater age
The authors are asked to explain the rationale for choosing the age range. Both vitamin and D levels were also analyzed with a method based on microparticles: the commercial specifications of the products used are required to be entered.
Scatter matrix graphs are presented but lack explanation
The axes and the reason why they were presented in this way are missing.
You are asked to rewrite the part and correct the graphs by adding the axes and explaining them better in the caption
When writing p it is required that p= be written and not just p because it is not clear
Table 2 and Table 3 need to be redone so that they are more usable by clearly placing values ​​and spacing
In the form presented they are not absolutely clear
It is required to better describe the population as it is defined as healthy but in reality, being at least half of the subjects affected by obesity, they cannot be defined as such.
you are asked to rewrite this statement clearly.
The text seems to be well written but the English could be improved.
There are no self-citations
The analyzes on SNPs are correct but need to be explained more clearly.
Article with MAJOR REVISION
Comments on the Quality of English Languagethe english can be improved
Author Response
Response to reviewers’ comments
Comments and Suggestions for Authors
The paper presented by Sater and colleagues explores the levels of monohydroxy vitamin D and its receptor and analyzes 6 different SNPs in healthy non-elderly-Bahraini-adults. The work is quite interesting but presents some critical issues
Note: Com. = Reviewer comment; Res. = Our response
The sample size is adequate and well differentiated between normal patients and patients with different degrees of obesity
Com. However, only patients with an age range of around 29.99±5.65 were selected as shown in table 1
Res. In this study, we aim to study the vitamin D system as the only possible cause of inflammation. The best approach to achieve this goal is to exclude all other possible causes of inflammation and the top of these causes of inflammation is aging, in addition, with increasing age also co-morbidities increase, although some may be sub-patent.
Com. The first question concerns the age of the population taken into consideration: vitamin D deficiency is usually associated with a greater age. The authors are asked to explain the rationale for choosing the age range.
Res. We believe the strength of this study is largely based on the study subject selection, by excluding subjects above 40 years, and subjects with patent disease, because both are possible causes of inflammation. In this study, we are not looking for the causes of vitamin D deficiency or its prevalence, but for the link between vit. D and inflammation. Worthy to mention that vitamin D deficiency is very frequent in our setting, almost 80%, on the contrary, we found a small age is more associated with D3 deficiency (Ref. PMID: 31595858). Therefore, we excluded subjects below 20 and above 40 years. Also, you can notice the low 25OHD3 levels in Table 2 in the study.
Com. Both vitamin and D levels were also analyzed with a method based on microparticles: the commercial specifications of the products used are required to be entered.
Res. Probably, you mean biomarkers and D3. The available information about the kit is: Chemiluminescent microparticle immunoassay (CMIA), Architect Abbott Diagnostics, Lake Forest, IL, USA). Unfortunately, currently, we don’t have the other details. As mentioned in the material and methods the analysis was done in SMC hospital, clinical biochemistry lab.
Com. Scatter matrix graphs are presented but lack explanation
You are asked to rewrite the part and correct the graphs by adding the axes and explaining them better in the caption. The axes and the reason why they were presented in this way are missing.
Res. Figure 1 is amended with the addition of the axis’s titles, and changes in the legend
Com. When writing p it is required that p= be written and not just p because it is not clear
Res. Corrections are done throughout including the figures, however, in several journals, typing of p-value is optional, and some journals use different style e.g., ‘p’ in italic
Com. Table 2 and Table 3 need to be redone so that they are more usable by clearly placing values ​​and spacing. In the form presented they are not absolutely clear
Res. We made some changes; however, the journal editorial team uses their own template, which we cannot argue.
Com. It is required to better describe the population as it is defined as healthy but in reality, being at least half of the subjects affected by obesity, they cannot be defined as such. you are asked to rewrite this statement clearly.
Res. We defined ‘healthy’ by adding extensions like; apparently healthy, asymptomatic, or clinically healthy, however, the term healthy is relative. ‘Healthy obesity’ is an accepted term by the WHO and many researchers, more precisely, it is described as metabolically healthy obesity (MHO), References include; [Blüher 2020, PMID: 32128581; Tsatsoulis, et al. PMID: 32301039, etc., in Pubmed]. Moreover, the subjects in this study might be the least unhealthy subjects compared to the ones described in most other studies on obesity.
New references are highlighted in the text and in the list in green and light green
Com. The text seems to be well written but the English could be improved.
Res. English is revised throughout the manuscript, using grammar software, and still subject to correction during proofreading
Com. There are no self-citations
Res. The citations are based on the suitability and relevance of the cited material, without bias; however, three of our articles are cited in this manuscript, (References 20, 22, and 28).
Com. The analyzes on SNPs are correct but need to be explained more clearly.
Res. We tried our best to clarify by limited editing in Table 3 and the text in the revised version of the manuscript.

Round 2
Reviewer 1 Report
Comments and Suggestions for Authors
1. The comment, “The methodology mentions that the sample collection is carried out in the Bahrain Defense Force (BDF) hospital in Bahrain. Authors should mention the type of sampling done and hence the bias associated with the study,” means that the authors should mention the Type of sampling, i.e., Simple Random, purposive, cluster, etc. This is important as this is a hospital-based study.
2. Comment No. 4 said, “As the inflammatory marker analysis is done using kits. Why have they mentioned the quantifications in arbitrary units and not in terms of concentrations (pg/ml)? Add the standard plot for these estimations.” The authors write, “This study aims to compare the groups and correlate variants for individuals, rather than to determine the actual values, which of no additional value, on the contrary, some values can read as zero after correction which does not suit the correlation analysis. Also, the outcome of the analysis will be the same, whether we used the arbitrary unit (AU), the corrected values, or the pg/ml, since the latter are derived from the AU using constant factors. Also ELISA is semi-quantitative and the results vary from lab to lab, kit to kit, setting to another. The pg/ml will be appropriate if cut-off level and reference ranges for diagnostic purposes are needed. However, the data and the standard plots are available if needed.” In that case, the results do not appear conclusive as only comparing groups with insignificant p values and hence, no correlation may fail to establish any concrete findings. I agree that the values vary from one lab to another and also from kit to kit, that is the reason that a standard plot is required, and details of the kit have to be mentioned. Kits also provide their working ranges.
Author Response
Comment: “The methodology mentions that the sample collection is carried out in the Bahrain Defense Force (BDF) hospital in Bahrain. Authors should mention the type of sampling done and hence the bias associated with the study,” means that the authors should mention the Type of sampling, i.e., Simple Random, purposive, cluster, etc. This is important as this is a hospital-based study.
Response: Thank you for this point. Although the sampling approach in this study appears to be random for studying vitamin D, it is a purposive/judgmental sampling because we selected a specific group of subjects for the study, which are the healthy non-elderly obese subjects. While this targeted approach inherently carries the risk of research bias, as the study design is based on predefined criteria that may align with the researchers’ hypotheses. However, we took careful steps to minimize potential bias by focusing on a relatively rare population—healthy, severely obese individuals.
Comment No. 4 said, “As the inflammatory marker analysis is done using kits. Why have they mentioned the quantifications in arbitrary units and not in terms of concentrations (pg/ml)? Add the standard plot for these estimations….
Response: Thank you for raising this important point regarding the standard plot. We have added the standard plots as a figure (Figure 1. Supplementary).
lthough the values of the presented markers are reported in pg/mL according to the kit’s instructions, we observed significant variations in the normal levels (healthy control subjects) of these biomarkers across different studies, including those on sepsis, Type 2 Diabetes (T2D), and obesity/vitamin D, even when conducted in our lab using the same kits and under similar conditions. As a result, we consider the reported values to be arbitrary units rather than true pg/mL concentrations. A review of several published articles confirmed similar discrepancies in the values of the same cytokines measured by ELISA. Given these variations, using pg/mL as the unit would be misleading. Furthermore, it is appropriate to use arbitrary units in this context, as our study is not diagnostic in nature.
All kits used in this study were purchased from Invitrogen (by Thermo Fisher Scientific), with the specific catalog numbers for each marker provided in the methodology section. The assay range for each marker is detailed in the respective kit's information sheet, which is available online on the Thermo Fisher website.

Reviewer 2 Report
Comments and Suggestions for Authors
The authors responded promptly to all comments and included new citations
The proposed graphics are much clearer and the English has been revised.
publication in this new form is recommended
Author Response
Comment: The authors responded promptly to all comments and included new citations The proposed graphics are much clearer and the English has been revised. publication in this new form is recommended
Response: Thank you very much for your thoughtful review of our manuscript and for recommending no further changes. We greatly appreciate the time and effort you dedicated to evaluating our work.
Round 3
Reviewer 1 Report
Comments and Suggestions for Authors
The authors have answered all my queries and incorporated most of the changes in the manuscript. The manuscript may be considered for publication.